# Biogenic Synthesis of Multifunctional Silver Oxide Nanoparticles (Ag_2_ONPs) Using *Parieteria alsinaefolia* Delile Aqueous Extract and Assessment of Their Diverse Biological Applications

**DOI:** 10.3390/microorganisms11041069

**Published:** 2023-04-20

**Authors:** Zakir Ullah, Farhat Gul, Javed Iqbal, Banzeer Ahsan Abbasi, Sobia Kanwal, Wadie Chalgham, Mohamed A. El-Sheikh, Sibel Emir Diltemiz, Tariq Mahmood

**Affiliations:** 1Department of Plant Sciences, Faculty of Biological Sciences, Quaid-i-Azam University, Islamabad 45320, Pakistan; 2Department of Botany, Bacha Khan University, Charsadda 24420, Pakistan; 3Department of Botany, Rawalpindi Women University, 6th Road, Satellite Town, Rawalpindi 46300, Pakistan; 4Department of Biology and Environmental Sciences, Allama Iqbal Open University, Islamabad 44000, Pakistan; 5Department of Mechanical and Aerospace Engineering, University of California, Los Angeles, CA 90095, USA; 6Botany and Microbiology Department, College of Science, King Saud University, Riyadh 11451, Saudi Arabia; 7Department of Chemistry, Eskisehir Technical University, Eskisehir 26470, Turkey

**Keywords:** *P. alsinaefolia*, Ag, SEM, XRD, DLS, zeta, cytotoxic, biocompatibility

## Abstract

Green nanotechnology has made the synthesis of nanoparticles a possible approach. Nanotechnology has a significant impact on several scientific domains and has diverse applications in different commercial areas. The current study aimed to develop a novel and green approach for the biosynthesis of silver oxide nanoparticles (Ag_2_ONPs) utilizing *Parieteria alsinaefolia* leaves extract as a reducing, stabilizing and capping agent. The change in color of the reaction mixture from light brown to reddish black determines the synthesis of Ag_2_ONPs. Further, different techniques were used to confirm the synthesis of Ag_2_ONPs, including UV-Visible spectroscopy, Fourier-transform infrared spectroscopy (FTIR), X-ray diffraction (XRD), scanning electron microscope (SEM), Energy-dispersive X-ray spectroscopy (EDX), zeta potential and dynamic light scattering (DLS) analyses. The Scherrer equation determined a mean crystallite size of ~22.23 nm for Ag_2_ONPs. Additionally, different in vitro biological activities have been investigated and determined significant therapeutic potentials. Radical scavenging DPPH assay (79.4%), reducing power assay (62.68 ± 1.77%) and total antioxidant capacity (87.5 ± 4.8%) were evaluated to assess the antioxidative potential of Ag_2_ONPs. The disc diffusion method was adopted to evaluate the antibacterial and antifungal potentials of Ag_2_ONPs using different concentrations (125–1000 μg/mL). Moreover, the brine shrimp cytotoxicity assay was investigated and the LC_50_ value was calculated as 2.21 μg/mL. The biocompatibility assay using red blood cells (<200 μg/mL) confirmed the biosafe and biocompatible nature of Ag_2_ONPs. Alpha-amylase inhibition assay was performed and reported 66% inhibition. In conclusion, currently synthesized Ag_2_ONPs have exhibited strong biological potential and proved as an attractive eco-friendly candidate. In the future, this preliminary research work will be a helpful source and will open new avenues in diverse fields, including the pharmaceutical, biomedical and pharmacological sectors.

## 1. Introduction

Nanotechnology is becoming a rapidly growing area of science and technology due to its wide range of applications in different industrial zones [1]. Nanotechnology deals with the synthesis of nanoparticles having a size range of 1–100 nm at least in one dimension. Nanoparticles (NPs) have significantly attracted the attention of the scientific community due to their remarkable features, such as small nanoscale size, a high percentage of surface atoms, particle structure, large surface-to-volume ratio, and unique shapes compared to their bulk counterparts [1,2,3,4]. Over the past two decades, several studies have been conducted in the fields of science, engineering, and biotechnology to create nanostructured materials that are affordable and eco-friendly [5,6,7,8]. Design, synthesis, and characterization of various metal nanoparticles for the treatment of multiple diseases have recently attracted the attention of nonscientists to combat various kinds of diseases [9,10,11]. They are used in many industrial fields because of their exceptional properties, including chemical, electrical, biomedical, and automotive industries [12,13]. This improves their ability to catalyze reactions and their ability to interact with other molecules [14,15]. Thus, different metal oxide nanoparticles (MNPs) have been synthesized. Among the different MNPs, silver oxide nanoparticles (Ag_2_ONPs) are one of the most popular types of metal oxide nanoparticles and have distinct uses. They have a large surface area despite their small size [16,17]. These nanoparticles are of great significance because of how their size affects the physiochemical characteristics of any material [18,19]. The MNPs are notable today in a variety of industrial applications of applied nanotechnology, widely used in a range of industries, including textile, food, health, agricultural, and cosmetics [20]. 

Nanoparticles are often produced via a range of physical and chemical techniques [21]. However, these conventional methods use expensive metal salts, organic solvents, and poisonous reducing stabilizing and capping agents and require expensive machinery which makes these synthesis routes very challenging. Further, these methods have several negative impacts on both the environment and human life, including cytotoxicity, carcinogenicity, and genotoxicity, which restricts their use for biomedical purposes [22]. These problems must thus be resolved, and steps must be taken to find a different way to make safe and biocompatible NPs. Consequently, researchers have created green chemical approaches. A novel and practical method in green synthesis is the biological production of nanoparticles. This procedure is highly simple, environmental-friendly, energy-efficient, non-toxic, and does not need a lot of pressure, heat, and energy. It also does not need any external chemicals for reducing, stabilizing, or capping metal precursors [23,24]. The type of plant extracts used, their concentration, and other factors significantly affect the physicochemical and biological potentials of eco-friendly nanoparticles [25]. Numerous beneficial phytochemicals found in natural plant extracts work as potent reducing, stabilizing, and capping agents in the synthesis of NPs [26,27]. The type of plant extracts used, extract concentration, metal salt, pH level, and synthesis method all affect how environmentally friendly nanoparticles are produced [28]. To produce less harmful chemical products and byproducts, researchers, chemical technologists, and chemists are currently using core principles of green chemistry as a reference manual for the green synthesis of MNPs across the globe [29,30]. Green nanotechnology has consequently witnessed significant growth as an alternative technique to produce safe and stable products using various medicinal plants [31,32,33].

The current study aimed to synthesize silver oxide nanoparticles (Ag_2_ONPs) using leaves extract of *P*. *alsinaefolia* Delile (*Urticaceae*), a versatile medicinal plant that can be utilized for a variety of treatments and is used in traditional medicine to treat intestinal worms, dysentery, diarrhea, and malarial fever [34]. The current work documents the first green synthesis of Ag_2_ONPs using leaves extract of *P*. *alsinaefolia* without the use of any surfactant or organic and inorganic solvent. Furthermore, the study was purposed to investigate numerous biological applications of a synthesized Ag_2_ONPs. 

## 2. Materials and Methods

### 2.1. Plant Extract Preparation

The medicinal plant *P*. *alsinaefolia* (*Urticaceae*) was collected from Rumli, Quaid-I-Azam University Islamabad, Pakistan. The plant sample was taxonomically identified by Dr. Sayed Afzal Shah, Assistant Professor (NUMS), Islamabad Pakistan, with authorization number: SAS-557. The leaf material was separated from *P*. *alsinaefolia*, carefully washed with deionized water, and dried in shade for 3 weeks so that the water content was removed completely. After that, the plant material was ground into a fine powder and stored in an air-tight container. Furthermore, 30 g of dried *P*. *alsinaefolia* powder was added to 250 mL of distilled water and heated at 80 °C. To obtain pure aqueous leaves extract, the solution was filtered 3 times using Whatman filter papers after being cooled to room temperature. The plant extract was then kept at 4 °C for future use.

### 2.2. Biosynthesis of Ag_2_ONPs

With few modifications, the procedure utilized by Iqbal et al. [35] was used to achieve the biosynthesis of Ag_2_ONPs. For this purpose, 3 g of silver nitrate salt (Sigma Aldrich, Saint Louis, MO, USA) was added to 100 mL of filtered *P*. *alsinaefolia* leaves extract and heated at 75 °C for 3 h with continuous stirring. Detail schematic representation of the study is provided in (Figure 1).

Moreover, the solution was centrifuged for 30 min at 3000 rpm. The resultant powder, which was assumed to be Ag_2_ONPs, was incubated for 5 h at 100 °C. To remove any impurities, the powder was gathered and rinsed thrice with distilled water followed by calcination to obtain pure-phase crystalline NPs. Furthermore, the NPs were extensively characterized using different characterization techniques. In addition, different in vitro biological activities were investigated for the synthesized NPs.

### 2.3. Characterization of Ag_2_ONPs

Using various characterization techniques, the type, structure, physical, chemical, and optical properties of Ag_2_ONPs have been investigated. Using an ultraviolet (UV-4000) UV-Vis spectrophotometer (Germany) with a wavelength range of 250–750 nm, the formation of Ag_2_ONPs was examined and validated. Fourier-transform infrared spectroscopy (FTIR) spectrometer was used to record the FT-IR spectra (Alpha, Bruker, Ettlingen, Germany). It was used to investigate the configuration and functional groups involved in the capping and efficient stabilization of Ag_2_ONPs. The samples were scanned utilizing a 500–4500 cm^−1^ spectral range to study the architecture and physical structure of Ag_2_ONPs. Further, Energy-dispersive X-ray spectroscopy (EDX) analysis was performed to determine the elemental composition of Ag_2_ONPs. The crystal clarity and particle size of Ag_2_ONPs were evaluated using a PANalytical Empyrean Diffractometer. The corresponding size of thermally annealed samples was determined using the Scherrer equation following analysis with an X-ray diffractometer equipped with a Cu radiation source at 45 kV and 40 mA current voltage. The sample was made by pouring Ag_2_ONPs powder onto a glass slide, air-drying it, and then using it for XRD examination. The dynamic light scattering (DLS) system explores the feasibility of the polydispersity index (PDI) and the hydrodynamic size distribution (Malvern Nano Zetasizer).

### 2.4. Bio-Potentials of Ag_2_ONPs

#### 2.4.1. Brine Shrimp Cytotoxicity Assay

Brine shrimp cytotoxicity assay was performed by putting the brine shrimp eggs in seawater, a sizable number of larvae were quickly hatched for experimental usage, making them the most practical test for toxicity investigations. To achieve this purpose, *Artemia salina* eggs were cultivated for 24 h at 27 °C in artificial seawater (3.8 g/L) in the presence of light to obtain mature nauplii. Roundabout 20 nauplii were put into a glass vial with seawater along with a sample and were analyzed by using a Pasteur pipette. Various Ag_2_ONPs dosages ranging from 37.5–1000 μg/mL were employed to determine their dose-dependent response. Glass vials containing seawater, vincristine sulfate, and mature nauplii were determined as positive controls, whereas in glass vials containing seawater, DMSO and mature nauplii were considered negative control. Further, the vials were incubated at 30 °C in an incubator for 24 h, then the number of dead shrimps was precisely counted in each vial. The LC_50_ values for Ag_2_ONPs were measured using GraphPad Prism version 8.0.0.

#### 2.4.2. Alpha-Amylase (AA) Inhibition Assay

The alpha-amylase inhibition assay was performed to determine the antidiabetic potential of Ag_2_ONPs. The reaction mixture was prepared by stepwise addition of using 25 μL of alpha-amylase, 40 μL of starch solution, 15 μL of phosphate-buffer saline (pH 6.8), and 30 μL of Ag_2_ONPs [36] into a microplate reader. All the components in the microplate were incubated for 90 min at 50 °C. After incubation, the mixture underwent the addition of 90 μL of iodine solution and 20 μL of 1 M HCL. Acarbose and Dimethyl sulfoxide (DMSO) was used as positive and negative control, respectively. A microplate reader was used to measure the optical density at 540 nm. GraphPad software was employed to calculate the median lethal concentration (LC_50_) value, using the formula below to determine percent inhibition:


% Inhibition = [Sample Absorbance − Absorbance of negative control]/[Absorbance of blank − Absorbance of negative control] × 100


#### 2.4.3. Antibacterial Activity of Ag_2_ONPs

The antibacterial activity of synthesized *P*. *alsinaefolia* Ag_2_ONPs was determined using the disc diffusion method using different bacterial strains. To achieve this purpose, strains of bacteria were subcultured overnight in nutrient broth media and then incubated at 37 °C for 24 h before the activity was performed. An overnight culture of various bacterial strains was distributed on pre-made agar media to assess the antibacterial potencies of Ag_2_ONPs. Filter discs loaded with various concentrations of Ag_2_ONPs (125–1000 μg/mL) were then dried and placed on top of the plates. The plates were monitored for ZOI for 24 h while they were incubated in an incubator at 37 °C. The antibiotic oxytetracycline was used as a positive control and 5% DMSO served as a negative control.

#### 2.4.4. Antifungal Activity

To examine the antifungal potential of Ag_2_ONPs, several strains of fungi were used by using autoclaved Sabouraud dextrose liquid media (SDA) [37]. Spores of fungi were subcultured in SDA and stored in an incubator for 24 h at 37 °C prior to the fungicidal assay. Various fungal strains liquid cultures had their optical density (OD) set to 0.5. Broth cultures were spread on an SDA medium using autoclaved cotton swabs. Different Ag_2_ONPs doses were used, ranging from 125 to 1000 µg/mL. Different doses of Ag_2_ONPs were used to examine their antifungal potentials. The Petri plates containing different fungal strains were incubated at 37 °C for 72 h to achieve this purpose. After 48–50 h mycelial growth was observed at various concentrations and hence ZOI was calculated. Amp-B was used as a positive control while DMSO was used as a negative control.

#### 2.4.5. Hemocompatibility Experiment Using Ag_2_ONPs

The hemocompatibility experiment was performed to find that Ag_2_ONPs are safe with human red blood cells (RBCs) to accomplish the biocompatibility experiment. To achieve this purpose, 90 µL of freshly acquired human RBC was put in an (EDTA) tube and spun at 10,000 rpm for 15 min. The pallet was then washed 2–3 times with PBS (phosphate buffer saline) (pH 7.4). Erythrocyte suspensions were made by combining 200 µL erythrocytes with 9.8 mL buffer (PBS), 100 µL of erythrocyte suspension was added to various Ag_2_ONPs concentrations and were incubated for 1–2 h at 37 °C and centrifuged at 12,000 rpm/15 min. Furthermore, the supernatant was transferred to a 96-well plate and hemoglobin release was measured at 540 nm using a microplate reader. The positive control used was Triton X-100, whereas the negative control was DMSO. The results are expressed as a percentage of hemolysis caused by various concentrations of Ag_2_ONPs and are determined using the formula below:


% Hemolysis = [Sample ABC − Negative Control]/[Negative control] × 100


#### 2.4.6. Antioxidant Capabilities of Ag_2_ONPs 

The approach of spectrophotometry was employed to assess the ability of silver oxide nanoparticles to scavenge free radicals. As a free radical, 2.4 mg 2,2-diphenyl-1-picrylhydrazyl (DPPH) was added to 25 mL methanol and appropriately vortexed. In the next step, we examined the free radical scavenging potential of Ag_2_ONPs at various concentrations ranging from 37.5–1000 μg/mL. Ascorbic acid (AA) was treated as a positive control while DMSO was used as a negative control. In addition, 20 µL of Ag_2_ONPs and 180 µL of reagent solution were added into a 96-well plate and incubated in dark for 1 h. The measurements were taken at 517 nm with a microplate reader. The DPPH scavenging percentage was calculated using the formula below:


% DPPH Scavenging = 1 − [Absorbance of the sample]/[absorbance of control] × 100


Furthermore, the total reducing power (TRP) of the test sample (Ag_2_ONPs) was analyzed using the earlier published Potassium-ferricyanide process [38,39,40]. The Ascorbic acid (AA) was taken as positive control while the DMSO served as a negative control. The optical density was measured at 630 nm using a microplate reader. The power reduction is shown in terms of gram ascorbic acid equivalent per mg (g AA/mg) of nanoparticles. The total antioxidant capacity (TAC) was also determined using the phosphomollybdenum standard technique [41]. The optical density was measured at 695 nm using a microplate reader. To compare the overall antioxidant potentials of Ag_2_ONPs, ascorbic acid was used as a positive control and DMSO as a negative control. The results are given in terms of ascorbic acid equivalents in micrograms per milligrams of the sample, or g AAE/mg.

## 3. Results and Discussion

### 3.1. Synthesis of Ag_2_ONPs

In the current study, *P*. *alsinaefolia* leaves extract was employed to quickly synthesize Ag_2_ONPs, acting as a bio-reductant and stabilizing agent. Ag_2_ONPs production has been optimized using *P*. *alsinaefolia* extract via the green method. It is a very significant medicinal plant with well-known therapeutic potential. Green synthesis is the most effective way when compared to physical and chemical methods [42]. Both synthesis processes, despite their efficiency, have some shortcomings, for example, high costs, high energy needs, and the creation of toxic hazardous waste materials [43]. Moreover, it is mentioned in previous work that some dangerous compounds might adhere to nanoparticles made by chemical methods, making them undesirable for diverse biological applications [44,45]. Consequently, the synthesis of Ag_2_ONPs using the green approach is favored because it has several features due to its ease of use, low cost, and non-use of toxic chemicals and solvents. Ag_2_ONPs formation was confirmed by a change in color from light brown to brownish black color shifts. This color change results from optical properties specifically, due to vibrations in surface plasmon. The biocompatible AgO nanoparticles were further studied using a wide range of characterization techniques, including UV, DLS, EDX, FT-IR, XRD, SEM, and ZETA.

### 3.2. UV-Vis Spectrophotometry

A color shift that indicates a successful reduction process was seen as the aqueous extracts were added to the precursor solution. A color change from light brown to darker brown shows an increase in bioreduction. Further, a UV-Vis spectrophotometer was used to scan the reaction mixture in ranges between 200 and 750 nm wavelength and the synthesis of AgO nanoparticles was confirmed. At 430–433, the absorbance of AgO was observed and these absorbance peaks fall inside the SPR range for Ag_2_ONPs. Because absorbance is inversely proportional to particle concentration, the reduction in absorbance indicates that the particles have settled. However, as no blue/red shift was seen at the peak point, we can attribute that NPs exhibit steady SPR behavior. Silver nanoparticles could be produced at concentrations as low as 1 mM AgNO_3_, but at higher concentrations the biosynthesis was negligible. To conduct more optimization trials, the concentration of 1 mM was processed. At 60 °C or above temperature, Ag nanoparticle production was noticed. Figure 2 indicates the biosynthesis peaked after three hours. The findings of our research utilizing UV-Vis spectroscopy are in accordance with those of an earlier study that used a variety of plant extracts [46,47,48].

### 3.3. X-ray Diffraction Spectroscopy (XRD)

The crystalline nature of silver nanoparticles was assessed by X-ray diffraction spectroscopy (XRD) analysis. The XRD pattern of Ag_2_ONPs produced during biosynthesis is shown in Figure 3. The acquired Bragg peaks were observed to be in accordance with crystallographic reflections from the JCPDS pattern 00-076-1393 at angles of 100 (27.43°), 110 (31.98°), 111 (46.89°), 200 (48.11°), 211 (54.3°), 211 (57.04°), 220 (68.72°), 310 (72.84°), and 311 (78.14°) (Table 1). Using Debye Scherrer’s equation (D = k/12 cos), the average size of Ag_2_ONPs was determined to be around 22.11 nm. These findings are consistent with earlier findings reported by [49,50,51]. Figure 3 illustrates the results of an X-ray diffraction investigation performed for annealed biogenic Ag_2_ONPs incubated at 100 °C. The single and pure phase of AgO was found to be in accordance with the observed Bragg peaks (JCPD card no: 079–1741). The absence of Bragg peaks for other closely similar substances demonstrates the pure crystalline nature of biogenic Ag_2_ONPs. The Ag_2_ONPs XRD pattern agrees with earlier reports.

### 3.4. Fourier Transform Infrared Spectroscopy (FTIR) 

The FT-IR study for AgONPs is shown in Figure 4. The evaluation of molecular vibrations and the presence of functional groups/ biomolecules essential to the efficient production and stabilization of AgONPs are determined using FT-IR spectra. CH3, C-C rocking and stretching bonds were stretched by the bands that appeared at 958.74/cm, mC-O-C bands at 1164.72 /cm and C(=O)-O stretching were stretched by the bands at 1228.56/cm. C-O stretches, vibrations of the aromatic ring, C-H stretches and O-H/C-H/N-H stretching of amines and amides are all visible in the bands at 1384.97, 1505.47, and 2924.41/cm, respectively. Further, the peak at 3369.31/cm indicates the stretching vibrations of amines and amides O-H/C-H/N-H. The Ag–O bond vibration was related to the bands that were observed at 521.64, 621.66, and 675.80/cm (Table 2).

### 3.5. Zeta Potential

The size and charge of Ag_2_ONPs mediated by *P*. *alsinaefolia* were evaluated using DLS and zeta potential. The Ag_2_ONPs displayed negative charge zeta potential, possibly because of the intense phytochemical adsorption on the synthesized NPs. They also increased their stability and prevent particles from aggregating. The polydispersity index was 0.522 (Figure 5A), and low PDI values suggested high-quality, polydisperse particles. These NPs were perfect for their biological activities due to the parameters. To be regarded as stable, NPs’ zeta potential values typically need to fall between +30 mV and −30 mV. The Zeta value for Ag_2_ONPs was found to be −18.5 mV (Figure 5B). An earlier study of Ag_2_ONPs utilizing *Ocimum basilicum* is in line with our results obtained [52].

### 3.6. Scanning Electron Microscopy (SEM)

The morphological makeup of the nanoparticles was determined using SEM examination. A small amount of the material was placed on a copper grid coated with carbon before being dried with a hand dryer to eliminate any extra particles. Using a scanning electron microscope, this produced grid was utilized to measure the size and form silver nanoparticles. Figure 6 shows SEM pictures of *P*. *alsinaefolia*-mediated Ag_2_ONPs and shows that synthesized NPs are spherical in shape with a mean crystal diameter of 22.11 nm, which was validated by the SEM investigation.

### 3.7. Energy-Dispersive X-ray Spectroscopy (EDX)

In addition to SEM, the EDX was employed to analyze the elemental composition of Ag_2_ONPs. EDX spectroscopy analysis was used to assess the elemental composition of silver nanoparticles. Analysis conducted using EDX has confirmed the existence of silver and oxygen. Figure 7 depicts the strong signals for oxygen and silver in the absence of any other signal, showing that synthesized Ag_2_ONPs are free of any other impurities. Strong indications for silver and oxygen were observed at 0.3, 3, and 3.1 KeV, and only the major elements “Ag” and “O”, which are related to the single-phase purity of the NPs have been identified in the EDX spectrum. When silver nitrate was in contact with *P*. *Alsinaefolia* leaf extract, the precursor silver nitrate basic was commonly reduced. In the leaves extract of *P*. *alsinaefolia*, many flavonoid chemicals have been discovered, including quercetin, Kaempferol-7-O methyl ether, emodin, gluside, Physicon 8 B-D gluside, and others [53,54,55]. These substances attach to the surface of metal ions and have a vital role in stabilizing NPs. They are present in the aqueous extract of *P*. *alsinaefolia*.

#### 3.7.1. Brine-Shrimp Cytotoxicity Test

To determine the cytotoxic capability of green Ag_2_ONPs against freshly hatched *A*. *salina*, the brine shrimp lethality assay (BSLA) was performed. The BLSA is the best assessment test to evaluate the cytotoxicity potential of any naturally occurring compound [56]. Ag_2_ONPs cytotoxic efficacy was investigated at various dosages between 37.5 and 1000 μg/mL. AgO nanoparticle concentrations of 1000 and 37.5 μg/mL, exhibited mortality rates of 100% and 20%, respectively. Our findings on silver-mediated particles are in line with earlier research on Ag_2_ONPs employing *Sargassum ilicifolium*, *Ocimum bacilicum*, *Sageretia thea*, and *Rhamnus virgata* [51,52,57,58]. Ag_2_ONPs capacity to cause cytotoxicity was investigated in a concentration-dependent manner and a dose-dependent response was observed, while their LC_50_ value (2.21 μg/mL) was measured. The cytotoxic potential increases as NP concentration increases (Figure 8). These findings supported Ag_2_ONPs capability to cause cytotoxicity. However, none of the Ag_2_ONPs doses assessed provided a higher percentage of inhibition than vincristine sulfate with LC_50_ (1.976 μg/mL) used as a (positive control).

#### 3.7.2. Alpha-Amylase Inhibition (AA) Assay

Diabetes mellitus is a condition where insulin, a hormone that is responsible for advancing fasting and postprandial blood glucose levels, is unable to appropriately regulate the homeostasis of lipid and carbohydrate metabolism [57]. Currently, the potential for AA inhibition of *P*. *alsinaefolia* mediated Ag_2_ONPs was investigated. The findings have examined how effective Ag_2_ONPs are at inhibiting AA. The considerable inhibitory potential of the Ag_2_ONPs ranges from 37.5 to 1000 μg/mL. As Ag_2_ONPs concentrations decrease, the rate of inhibition gradually slows down. At 1000 μg/mL, maximum inhibition of 66% was observed. Decreased results were seen with 37.9% at 500 μg/mL and 20.83% at 250 μg/mL, respectively. However, none of the Ag_2_ONPs doses that were examined provided an inhibition percentage greater than (surfactin), which is a positive control. Since the AA enzyme acts by converting carbohydrates into glucose [58], inhibiting its activity might lower blood sugar levels, which is a significant subject of inquiry in diabetes [59]. Figure 9 indicates the AA inhibition potency of Ag_2_ONPs. Our results of green Ag_2_ONPs substantiate through the *S. thea*, *R. virgata*, *Calendula officinalis*, and mediated Ag_2_ONPs [52,58,60].

#### 3.7.3. Antibacterial Activity

Therapeutic approaches frequently provide the option of antibiotic treatment for bacterial infections; however, medications accompany downsides, such as antibiotic resistance. The scientific community is putting a lot of effort into creating novel methods to prevent antibiotic resistance and decrease the spread of these deadly diseases [61,62,63]. Promising new techniques in nanotechnology are available to design and produce novel materials with unique antibacterial characteristics [64]. The focus has shifted to finding new ways to solve these issues, resulting in the design of green synthesized nanoparticles (NPs). Silver nanoparticles can release silver ions that can pass through the cell membrane of microorganisms [65]. As a result, organelles functioning may be disturbed due to lysis of cytoplasmic membrane and cell lysis may even follow.

Currently, silver nanoparticles were evaluated for antibacterial activity. The antibacterial potential of biogenic Ag_2_ONPs, together with various bacterial strains at different dosages (1000–125 μg/mL), was assessed and their results are shown in Figure 10
*Staphylococcus aureus* (ATCC 23235), *Lactobacillus acidophilus* (ATCC 4356), and *Bacillus subtilis* (ATCC 23857) were used as gram-positive bacteria strains, whereas *Pseudomonas aeruginosa* (ATCC 15442), and *Escherichia coli* (ATCC BAA-2471) were used as gram-negative bacteria strains. Our silver-mediated particles showed dose-dependent responses against selected bacterial strains. Most studied microorganisms were found to be sensitive to AgO_2_NPs in this study. Our investigation revealed that the *S. aureus* bacterial strain was more vulnerable to silver-mediated particles at a concentration of 1000 μg/mL with 23.25 mm ZOI and at 125 μg/mL with 6.5 mm ZOI, respectively. Similarly, *L*. *acidophilus* shows 23 mm ZOI at 1000 μg/mL and 7 mm at 125 μg/mL. In the case of *E*. *coli* at 1000 μg/mL, ZOI was 22.25 mm and at 125 μg/mL, it was 8.5 mm. Moreover, *Pseudomonas aeruginosa* gives 22.75 mm of ZOI at 1000 μg/mL, and 6.85 mm at 125 μg/mL, respectively. Furthermore, the *Bacillus subtilus* is somehow susceptible to Ag_2_ONPs and its ZOI at 1000 μg/mL was 22.5 mm, while it shows 7 mm at 125 μg/mL. Table 3 lists the ZOI values, with oxytetracycline (10mg) as a positive control. The oxytetracycline was revealed to be more effective than any single test sample concentration. Overall, the ZOI of various strains is presented in Figure 11. Generally, we reported probable biogenic Ag_2_ONPs antibacterial activity that is in line with prior research [58,64,66]. The bioactive functional groups associated with NPs may be the cause of their potential antibacterial activities. Comparably, our research found that an increase in the concentration of *P. alsinaefolia*-mediated Ag_2_ONPs was correlated with an enhancement in antibacterial potential. In addition to ROS production, other variables, such as membrane damage from NPs adhering to the surface, might harm cells. The antibacterial potential of NPs can also be described by surface imperfections in the symmetry, which can harm cells [67]. We also consider the importance of bioactive functional groups that are connected to *P*. *alsinaefolia* leaves in the aqueous extract, which is employed to cap and stabilize Ag_2_ONPs and have significant antibacterial activity. 

#### 3.7.4. Antifungal Assay

The antifungal activity of *P*. *alsinaefolia*-mediated Ag_2_ONPs was evaluated against various fungal strains at different doses, i.e., (1000–125 μg/mL). The fungal strains assessed against Ag_2_ONPs were *Mucor racemosus* (FCBP 0300), *Aspergillus flavus* (FCBP: 0064), *Aspergillus niger* (FCBP: 0918), *Candida albicans* (FCBP: 478), and *Fusarium solani* (FCBP: 0291). Ag_2_ONPs against certain fungal strains have been the subject of extensive research. The present study marks the first to describe the antifungal activity of Ag_2_ONPs inspired by *P*. *alsinaefolia*. 

In the recent study, the susceptibility of the chosen strains was calculated by measuring the mycelial growth on SDA media at various Ag_2_ONPs concentrations (1000–125 μg/mL). Following the work, *Aspergillus flavus* was identified with a ZOI of 27.5 ± 0.71 mm at 1000 μg/mL and 9.5 ± 1.4 mm at 125 μg/mL. Likewise, at 1000 μg/mL, *Aspergillus niger* exhibits a ZOI of 29.5 ± 0.71 mm, *Candida albicans* are constrained to a 29 ± 1.41 mm region, while *M*. *racemosus* strain exhibits a 30.5 ± 0.71 mm ZOI, respectively (Figure 12). However, none of the test samples demonstrated% inhibition greater than *Amp-B*. According to earlier research, Ag_2_ONPs interact with fungal hype and spores, which inhibits fungal growth in addition to generating ROS [68]. Previous investigations [69] demonstrated significant dose-dependent antifungal activity that is compatible with the present findings. Figure 13 presents the overall ZOI studied fungal strains. Comparably, our study found that an increase in the concentration of *P. alsinaefolia*-mediated Ag_2_ONPs was associated with a boost in antifungal potential.

#### 3.7.5. Antihemolytic Potential of P.A-Mediated Ag_2_ONPs on Human RBCs

An antihemolytic assay was carried out using human RBCs to confirm the biosafety of the substance. Human RBCs were used to evaluate the biocompatibility and toxicological impact of silver oxide nanoparticles. A biological substance is considered hemolytic if it has an activity level of at least 5%, slightly hemolytic if it is between 2–5%, and non-hemolytic if it is less than 2% [70]. If a particular nanoparticle is hemolytic, it will shatter red blood cells and release hemoglobin. Currently, the RBCs were subjected to Ag_2_ONPs in concentrations ranging from 1000 to 75 µg/mL. The data collected showed that the synthesized NPs were non-hemolytic at lower concentrations (17, 35, and 71 µg/mL), somewhat hemolytic at 75–125 µg/mL, and hemolytic at concentrations of >125 µg/mL These findings support the earlier studies of *Cichorium intybus* and *Annona muricata* mediated Ag_2_ONPs [71,72]. Our study provided evidence that the biosynthesized Ag_2_ONPs are non-hemolytic and regarded as biocompatible at low concentrations. Figure 14 summarizes the results of Ag_2_ONPs biocompatibility assays.

#### 3.7.6. Antioxidant Activities

A cell under stress may produce ROS because of oxidative chain reactions. One or more unpaired electrons make up free radicals which are separate chemical compounds that are extremely unstable for the reason they break down neighboring molecules by removing electrons to become stable [73]. To quench these free radicals and assist the cells in regaining normal function, antioxidants play a crucial role. Various green oxide metal nanoparticles have shown significant antioxidant results. Currently, the antioxidant activities of Ag_2_ONPs (TAC, TRP, and DPPH free radical scavenging) are being investigated (Figure 15). An aqueous extract of *P*. *alsinaefolia* leaves was employed as a capping, reducing, and oxidizing agent. Several phenolic compounds are inferred to scavenge the ROS which is also coupled to the Ag_2_ONPs. The range of dilution concentrations was 75–1000 μg/mL for the antioxidant analyses. The highest result for total antioxidants was determined to be 51.4% for 200 μg/mL of Ag_2_ONPs in terms of AA equivalents per mg. The lowest value for TAC activity was 6.4 ± 4.7, while the highest value was 87.5 ± 4.8 at a concentration of 1000 μg/mL. TAC (Figure 15A) shows the scavenging potential of the tested compounds toward ROS species.

Total reducing power (TRP) was studied to further examine the presence of antioxidant species adsorbed to Ag_2_ONPs. This technique was carried out to look at reductones that contribute H-atoms to the antioxidant potential which may cause chain damage from free radicals. Biogenic Ag_2_ONPs have demonstrated significant antioxidant capacity. As the concentration of Ag_2_ONPs decreased, the reducing power also lowered. At 60 μg/mL, the greatest reducing value was determined to be 62.68 ± 1.77%, while the lowest value was 13.88 ± 1.36% at 5 μg/mL (Figure 15B). Additionally, at 200 μg/mL, Ag_2_ONPs showed a significant DPPH radical scavenging capacity (79.4%). According to the findings in Figure 15C, it is possible that several antioxidant substances contribute to the decrease and stability of Ag_2_ONPs by *P*. *alsinaefolia* leaves extract. Our findings agree with previous investigations of biogenic Ag_2_ONPs utilizing *Citrus limon*, and *Cassia Auriculata* [74,75].

## 4. Conclusions

Currently available biogenic Ag_2_ONPs were produced using *P*. *alsinaefolia* leaf extract and were thoroughly examined using XRD, FTIR, EDS, SEM, DLS, and zeta potential. Biogenic Ag_2_ONPs have also demonstrated diverse in vitro biological activity. The antibacterial and antifungal properties of the biogenic Ag_2_ONPs have been intriguing. Additionally, assessments for enzyme inhibition and modest antioxidant activity have been observed. We studied into Ag_2_ONPs biosafety in relation to RBCs in more depth. Green synthesis is the route to go for producing NPs that might be utilized for the detection and treatment of many diseases while avoiding the usage of hazardous chemicals. In addition, the application of green Ag_2_ONPs in biomedicine is a vast field that needs in-depth study.

## Figures and Tables

**Figure 1 microorganisms-11-01069-f001:**
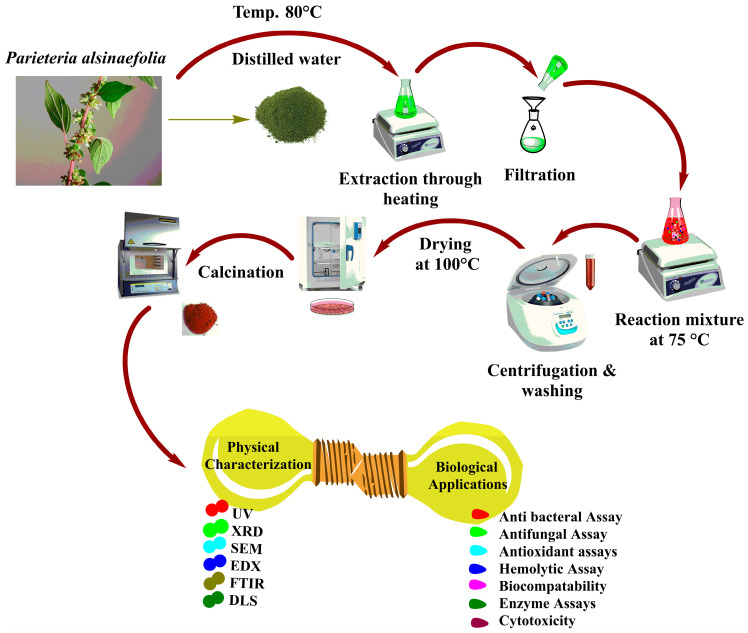
The detailed mechanism of *P*. *alsinaefolia* mediated Ag_2_ONPs synthesis.

**Figure 2 microorganisms-11-01069-f002:**
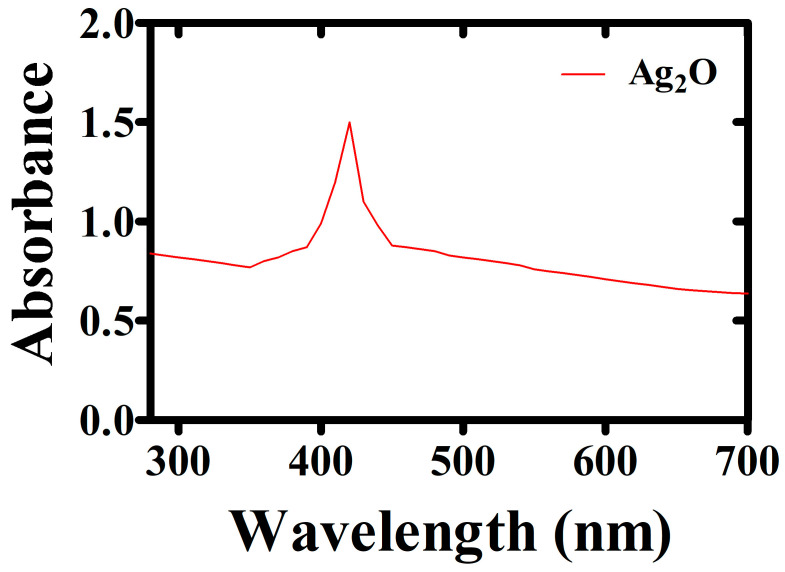
Biologically produced calcinated Ag_2_ONPs UV-visible spectrum.

**Figure 3 microorganisms-11-01069-f003:**
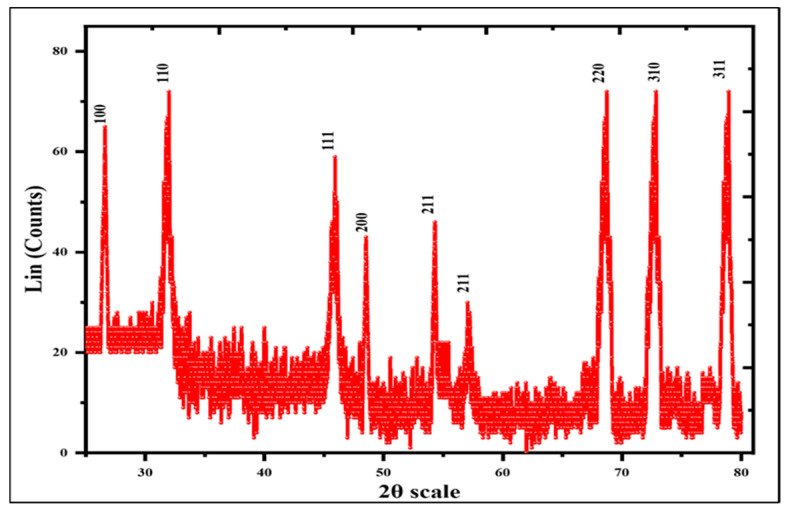
XRD spectra of Ag_2_ONPs developed by *P*. *alsinaefolia*.

**Figure 4 microorganisms-11-01069-f004:**
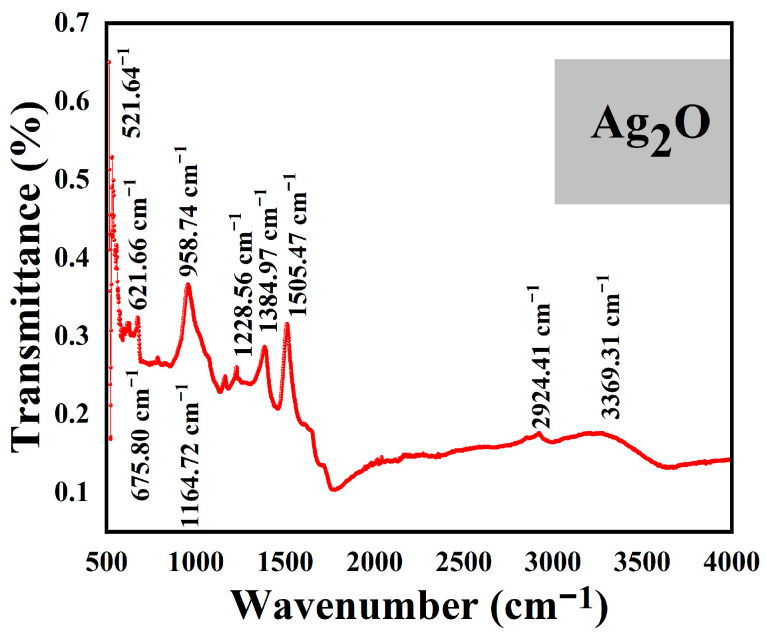
FT-IR spectra of *P*. *alsinaefolia* Ag_2_ONPs determining different functional groups.

**Figure 5 microorganisms-11-01069-f005:**
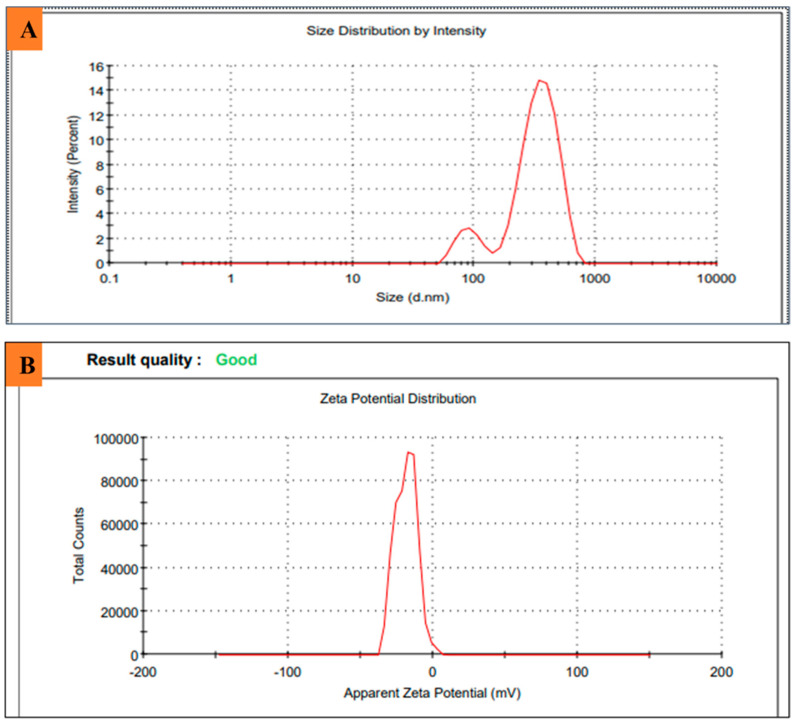
(**A**) Particle size distribution of *P*. *alsinaefolia* Ag_2_ONPs; (**B**) zeta potential measurement of *P*. *alsinaefolia* Ag_2_ONPs.

**Figure 6 microorganisms-11-01069-f006:**
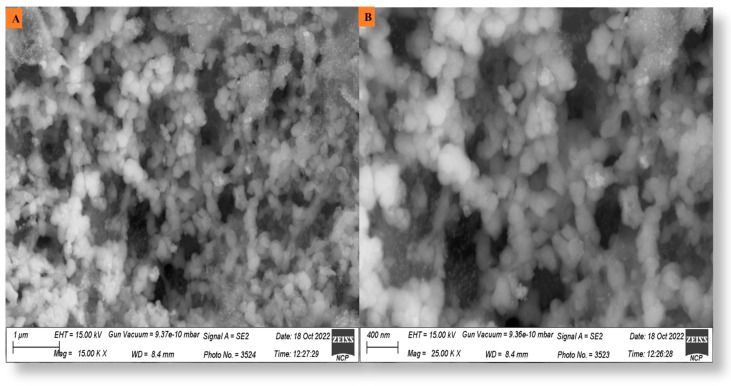
(**A**) and (**B**) SEM analysis of biogenic *P*. *alsinaefolia*-mediated Ag_2_ONPs determining the topology of Ag_2_ONPs.

**Figure 7 microorganisms-11-01069-f007:**
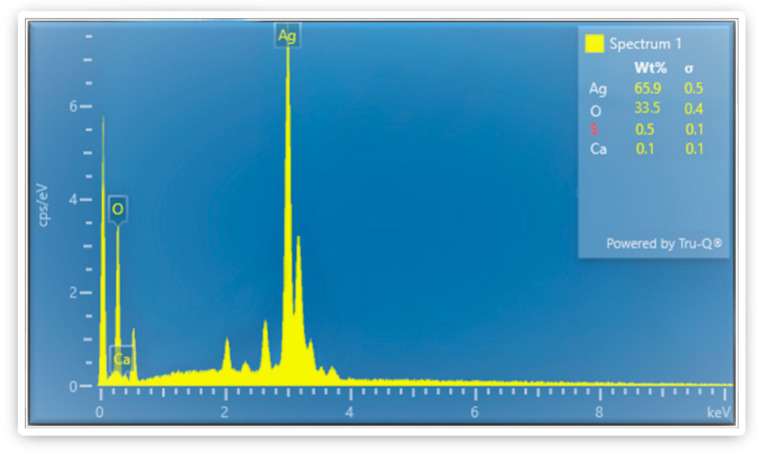
Elemental composition of Ag_2_ONPs using EDX.

**Figure 8 microorganisms-11-01069-f008:**
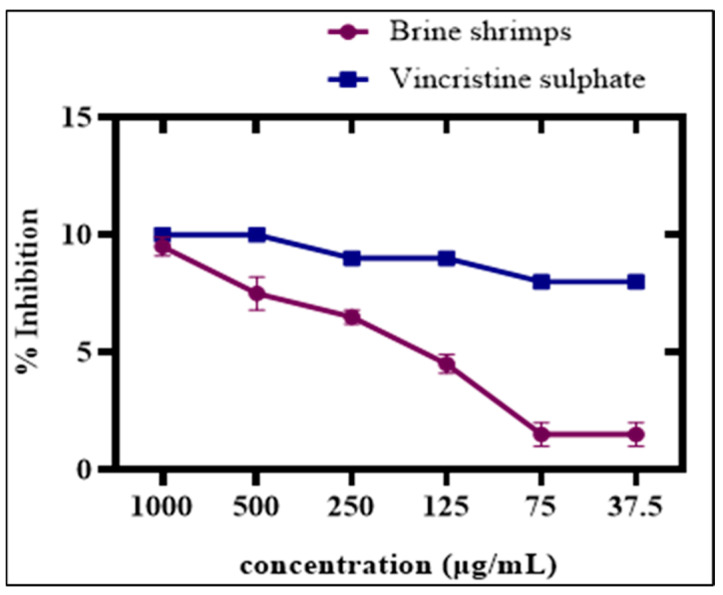
Cytotoxic effects of P.A mediated Ag_2_ONPs on brine shrimp.

**Figure 9 microorganisms-11-01069-f009:**
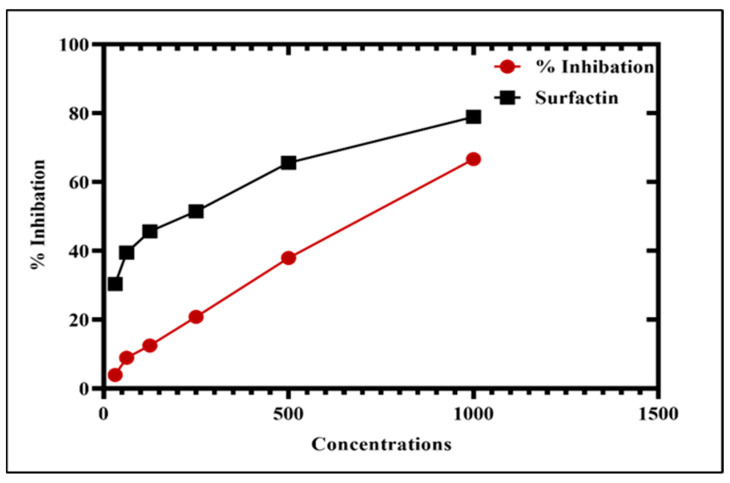
Antidiabetic potential of Ag_2_ONPs against alpha-amylase.

**Figure 10 microorganisms-11-01069-f010:**
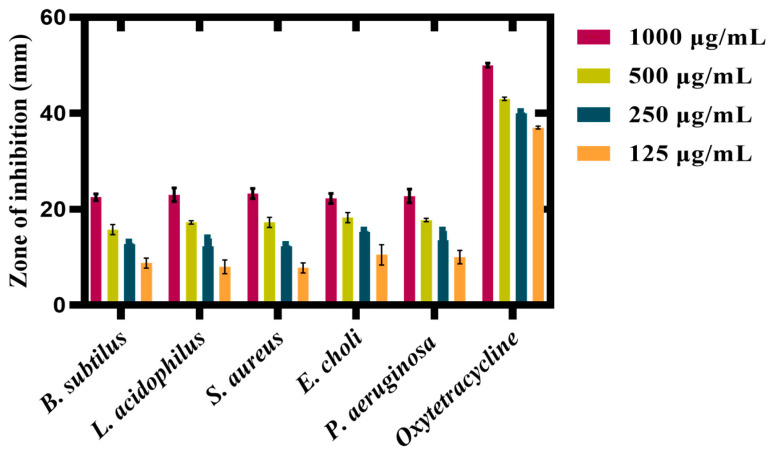
Antibacterial potential of mediated Ag_2_ONPs.

**Figure 11 microorganisms-11-01069-f011:**
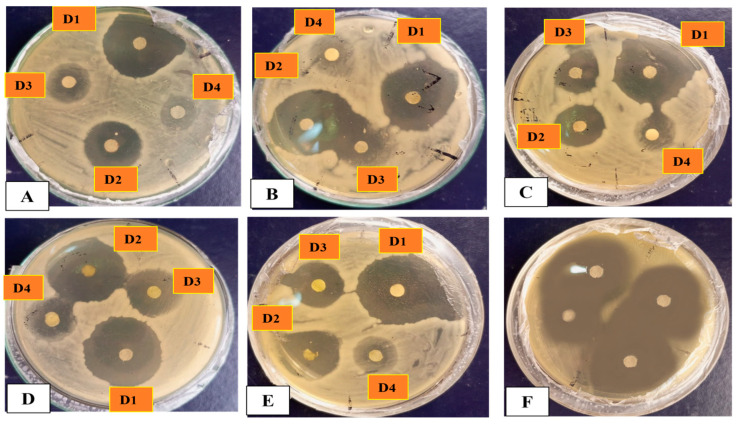
The zone of inhibition with different dilution (D1–D4) (1000-125 μg/mL) of various bacterial strains. (**A**) *B. subtilus*, (**B**) *L. acidophilus*, (**C**) *S. aureus*, (**D**) *E. coli*, (**E**) *P. aeruginosa*, (**F**) control *(Oxytetracycline)* against Ag_2_ONPs with different concentrations.

**Figure 12 microorganisms-11-01069-f012:**
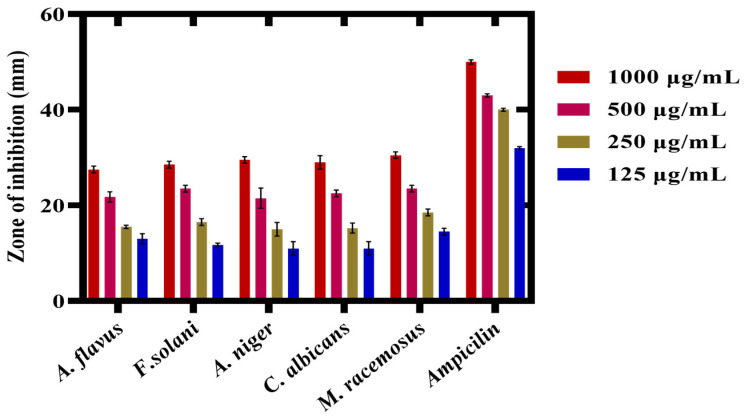
Antifungal potential of biogenic Ag_2_ONPs.

**Figure 13 microorganisms-11-01069-f013:**
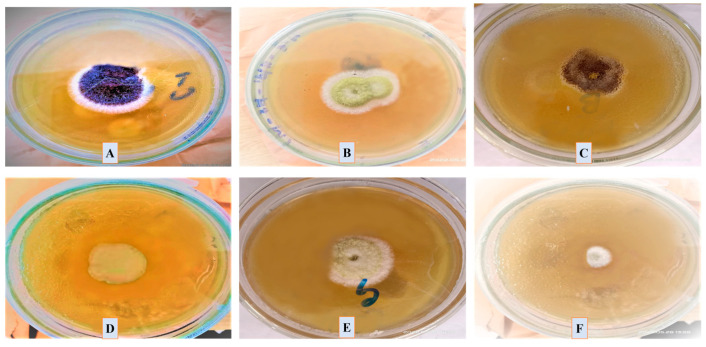
Mycelial growth and Zone of inhibition of various fungal strains. (**A**) *A. flavus*, (**B**) *F. solani*, (**C**) *A. niger*, (**D**) *C. albicans*, (**E**) *M. racemosus*, (**F**) control (*Ampicillin*) against Ag_2_ONPs.

**Figure 14 microorganisms-11-01069-f014:**
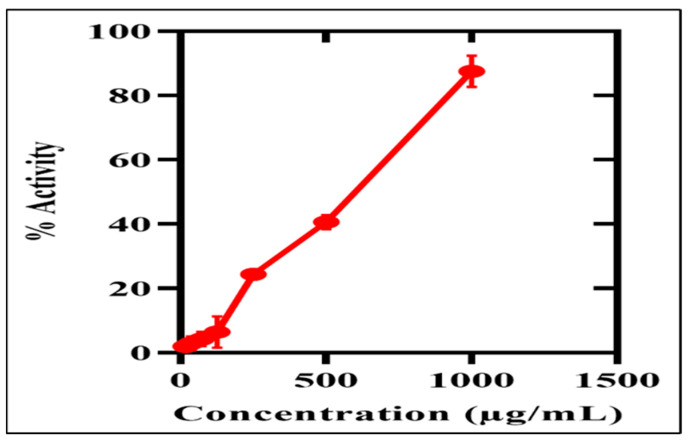
The antihemolytic potential of *P. alsinaefolia*-mediated Ag_2_ONPs.

**Figure 15 microorganisms-11-01069-f015:**
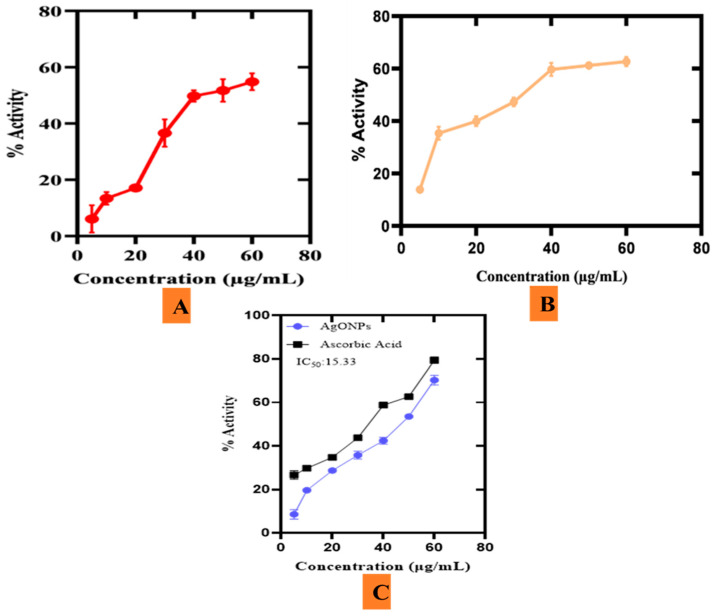
(**A**) Antioxidant (TAC) potential of *P*. *alsinaefolia*-Ag_2_ONPs. (**B**) Antioxidant (TRP) potential of *P*. *alsinaefolia*-Ag_2_ONPs. (**C**) Antioxidant (DPPH) potential of PA-Ag_2_ONPs.

**Table 1 microorganisms-11-01069-t001:** (2 thetas) value and Bragg peak of *P*. *alsinaefolia* Ag_2_ONPs.

S.No	2 Theta Value	Bragg Peak
1	27.43	100
2	31.98	110
3	46.89	111
4	48.11	200
5	54.3	211
6	57.04	211
7	68.72	220
8	72.84	310
9	78.14	311

**Table 2 microorganisms-11-01069-t002:** Functional group that linked with the surface area of the *P. alsinaefolia*-Ag_2_ONPs.

S.No.	Wavenumber (cm^−1^)	Functional Groups
1	3369.31	O-H/C-H/N-H stretching of amines and amides
2	2924.41	C-H
3	1505.47	Vibrations of the aromatic ring
4	1384.97	C-O
5	1228.56	C(=O)-O stretching
6	1164.72	mC-O-C
7	958.74	CH3, C-C rocking and stretching bonds
8	521.64, 621.66 and 675.80	Ag-O bond vibrations

**Table 3 microorganisms-11-01069-t003:** ZOI value of different bacterial strains.

Bacterial Strains	ZOI (mm)
*B. subtilus*	22.5
*L. acidophilus*	23
*S. aureus*	23.25
*E. coli*	22.25
*P. aeruginosa*	22.75
*Oxytetracycline*	50

## Data Availability

All the raw data of this research can be obtained from the corresponding authors upon reasonable request.

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
