# Peer review of "Biogenic Synthesis of Multifunctional Silver Oxide Nanoparticles (Ag2ONPs) Using Parieteria alsinaefolia Delile Aqueous Extract and Assessment of Their Diverse Biological Applications"

_microorganisms, 2023, doi:10.3390/microorganisms11041069_

Round 1

Reviewer 1 Report

Reviewer comments

I have gone through the manuscript titled “Biogenic synthesis of multifunctional Silver Oxide Nanoparticles Using Parieteria alsinaefolia Aqueous Extract and assessment of Their Diverse Biological Applications".  The work describes a Parieteria alsinaefolia extract functionalized green synthesis of AgONPs and evaluated their multiple in vitro bioactivities including antibacterial, antifungal, antioxidant, cytotoxicity etc. to determine their biological and biocompatibility potentials. The study is interesting in terms of its characterizations and biomedical applications. Manuscript is well written and results have been justified in an excellent way. The manuscript is simple, well-organized and easy to read. However, before the paper can published/ accepted, it is necessary for authors to undertake minor revisions in accordance with comments as suggested by me.

Comment: Please give the abbreviation of (AgONPs) along with the full name in the title of the manuscript.

Comment: Authority citation is missing in front of medicinal plant used. Provide authority citation.

Comment: Replace the following words with as suggested.

Replace the word “a” with “has” in the statement “Nanotechnology has a significant impact………..versatile applications in different commercial areas”.

Replace the word “further” with “furthermore” in throughout the manuscript wherever needed in the manuscript.

Comment: The experimental part: “A well-known medicinal plant Parieteria alsinaefolia was collected and taxonomically identified. The introduction needs to build up to the reasons why, is this plant useful. Why the authors have selected this plant, mention on the introduction section.

Comment: There are some typographic errors in the introduction, carefully review manuscript and remove errors where needed.

Comment: In the introduction and discussion:  All of the scientific names in main text should be italicized.

Comment: Authors should also expand the abbreviations used in the abstract.

Comment: Authors have cited figures in the text with different formats, please check for consistency.

Comment: The authors should italicize the name of the family of plant used. In addition, the authors should provide authority citation for the plant used.

Comment: Don’t start sentence directly with “AgONPs” use full form or use the word “The” before AgONPs or type it as silver nanoparticle.

Comment: Improve figure captions and explain it well.

Author Response

Thank you so much for reviewing our manuscript by providing your quality time, valuable comments/ recommendations. Your comments has really improved the quality and structure of our manuscript. Our team members have extensively and carefully reviewed the manuscript and properly addressed all suggestions/recommendations point by point as suggested by worthy reviewers. Now, we hope that the revised manuscript is highly improved. We have again critically reviewed the manuscript for typos errors and other mistakes. All necessary changes made have been highlighted yellow. If you recommend further suggestions, let us know, we will be very happy to address.

REVIEWER# 1

Comment: Please give the abbreviation of (AgONPs) along with the full name in the title of the manuscript.

Response: The abbreviation AgONPs has been provided in front of silver oxide nanoparticles in the title.

Comment: Authority citation is missing in front of medicinal plant used. Provide authority citation.

Response:  The valid authority citation has been provided in the revised manuscript and provided with yellow color.  

Comment: Replace the following words with as suggested.

Comment: Replace the word “a” with “has” in the statement “Nanotechnology has a significant impact………..versatile applications in different commercial areas”.

Response: The word “a” has been replaced with “has” in the statement “Nanotechnology has a significant impact………..versatile applications in different commercial areas.

Comment: Replace the word “further” with “furthermore” in throughout the manuscript wherever needed in the manuscript.

Response: Replace the word “further” with “furthermore” in the revised manuscript wherever needed and highlighted yellow.

Comment: The experimental part: “A well-known medicinal plant Parieteria alsinaefolia was collected and taxonomically identified. The introduction needs to build up to the reasons why, is this plant useful. Why the authors have selected this plant, mention on the introduction section.

Response: I have provided justification for why the medicinal plant Parieteria alsinaefolia was selected for the green synthesis of nanoparticles. As the plants contain different bioactive phytochemicals and research studies have shown that it is used for the treat intestinal worms, dysentery, diarrhea, and malarial fever etc.

Comment: There are some typographic errors in the introduction, carefully review manuscript and remove errors where needed.

Response: We have extensively reviewed the manuscript and corrected typographic errors wherever needed.

Comment: In the introduction and discussion:  All of the scientific names in main text should be italicized.

Response: The medicinal plant names have been reviewed and italicized in the revised manuscript and highlighted yellow.

Comment: Authors should also expand the abbreviations used in the abstract.

Response: The abbreviation used have been now provided with their full forms in the abstract.

Comment: Authors have cited figures in the text with different formats, please check for consistency.

Response: The different figure have been reviewed for the said comment and addressed wherever needed.

Comment: The authors should italicize the name of the family of plant used. In addition, the authors should provide authority citation for the plant used.

Response: The family name of the targated plant used have been italicized and highlighted yellow.

Comment: Don’t start sentence directly with “AgONPs” use full form or use the word “The” before AgONPs or type it as silver nanoparticle.

Response: The manuscript has been carefully reviewed for the said comment and addressed. Sentences directly starting with “AgONPs” have been corrected.

Comment: Improve figure captions and explain it well.

Response: The figure captions have been reviewed and improved and changes made have been highlighted yellow.

Reviewer 2 Report

In this study, authors reported Biogenic synthesis of multifunctional Silver Oxide Nanoparticles Using Parieteria alsinaefolia Aqueous Extract and assessment of Their Diverse Biological Applications. However, the manuscript contains many flaws that should be addressed.

Some critical comments and suggestions

1.      Figure 1. AgONPsNanoparticles should be corrected.

2.      Figure 2 should be deleted from the main manuscript. It can be presented in the supplementary section.

3.      Figure 3 is not considered for scientific publication.

4.      Figure 4, the crystal spectrum does not correlate to the AgO. Moreover, the provided JCPD no not for Ag. The author should be explained.

5.      Figure 5 should be presented appropriately. For example, the absorbance spectra are wrongly mentioned as the transmittance and wavenumber should be corrected.

6.      Figure 7 should be presented with a scale bar.

7.      How the AgO formed without applying the oxidation process. The further author should evidence of the existence of Ag and O by elemental mapping.

8.      Figure 11 is not scientific and does not need a pictorial representation.

9.      Antimicrobial activity should be given in the evidential photograph of the plates in the main or supplementary section.

10.  The presented results were not organized well.

Author Response

Thank you so much for reviewing our manuscript by providing your quality time, valuable comments/ recommendations. Your comments has really improved the quality and structure of our manuscript. Our team members have extensively and carefully reviewed the manuscript and properly addressed all suggestions/recommendations point by point as suggested by worthy reviewers. Now, we hope that the revised manuscript is highly improved. We have again critically reviewed the manuscript for typos errors and other mistakes. All necessary changes made have been highlighted yellow. If you recommend further suggestions, let us know, we will be very happy to address.

Comment: Figure 1. AgONPs Nanoparticles should be corrected.

Response: Thanks for comprehensive review and highlighting the mistakes. The purpose of AgONPs was to abbreviate silver oxide nanoparticles. Now we have corrected the abbreviation as suggested. “AgONPs” is replaced with “Ag2ONPs” in Figure 1 and rest of the manuscript.

Comment: Figure 2 should be deleted from the main manuscript. It can be presented in the supplementary section.

Response: Figure 2 is now deleted from the revised manuscript and added to supplementary data by S-1 as suggested.

Comment: Figure 4, the crystal spectrum does not correlate to the AgO. Moreover, the provided JCPD no not for Ag. The author should be explained.

Response: Thank you so much for highlighting the typos error. We have corrected the typos error, added correct JCPD card No. and highlighted yellow in the revise manuscript.

Comment: Figure 5 should be presented appropriately. For example, the absorbance spectra are wrongly mentioned as the transmittance and wavenumber should be corrected.

Response: Thank you so much for taking interest in reviewing our manuscript and correction. The absorbance spectra are mentioned as the transmittance and wavenumber have been corrected and provided with revised figure.

Comment: Figure 7 should be presented with a scale bar.

Response: Figure 7 now figure 6 in the revised manuscript is provided with scale bar.

Comment: How the AgO formed without applying the oxidation process. The further author should evidence of the existence of Ag and O by elemental mapping.

Response: Thank you so much for your comprehensive review. Highly appreciated. Synthesis of AgO NPs can takes place without applying oxidation process by running reaction in water and open atmosphere .Because,  During the Ag Nps formation,  it's near about  impossible to stop the surface oxidation of Ag due to the high surface energy of Ag in the oxygenated environment. The elemental mapping is evident from figure 7 (revised manuscript).

Comment: Figure 11 is not scientific and does not need a pictorial representation.

Response: Figure 11 has been deleted from the main manuscript as suggested.

Comment: Antimicrobial activity should be given in the evidential photograph of the plates in the main or supplementary section.

Response: Evidential photographs are now added to the revised manuscript. Thanks for improving the structure and quality of our articles by providing your valuable comments.

Comment: The presented results were not organized well.

Response: All authors have carefully reviewed the results, made necessary changes wherever need and we hope that our results are well-organized and presented more scientifically..

Round 2

Reviewer 2 Report

Received the manuscript “Biogenic synthesis of multifunctional Silver Oxide Nanoparticles (Ag2ONPs) using Parieteria alsinaefolia Delile Aqueous Extract and assessment of Their Diverse Biological Applications” for review.  Though I received a revised manuscript, not addressed previous comments appropriately.

Some critical comments and suggestions

1.      What author does mean by the first line of the abstract?

2.      In the first version authors mentioned AgONPs but then how corrected into “Ag2ONPs”. This should be evidenced through XPS analysis.

3.      Figure 1 was not aligned appropriately and not presented according to the methods.

4.      Figure 4 is still not correct.

5.      The authors provided the elemental spectrum but not elemental mapping.

6.      Figures 10 and 11 are not correlated with each other.

7.      Antifungal activity should be given in the evidential photograph of the plates in the main or supplementary section.

8.      The manuscript contains many flaws in terms of scientific content.

Author Response

Thank you so much for reviewing the manuscript by providing your valuable comments and recommendations that has really improved the quality and structure of our manuscript. All suggestions/recommendations have been properly addressed as the reviewers suggested/asked and now we hope that the manuscript is highly improved. We again critically reviewed the manuscript for typos errors and other mistakes. If you recommend further suggestions, let us know, we will be very happy to address.

Comment: What author does mean by the first line of the abstract?

Response: Thank you so much for your review. The first line is presented scientifically in the revised manuscript.

Comment: In the first version authors mentioned AgONPs but then how corrected into “Ag2ONPs”. This should be evidenced through XPS analysis.

Response: Thanks for highlighting the error. This issue has been addressed throughout the manuscript. Please accept our apology, AgONPs was typographical error. The asynthesized nanoparticles are Ag2ONPs. We have already confirm the synthesis of Ag2ONPs using different characterization techniques. I have asked and inquired from multiple research centers and universities to perform this XPS analysis, but we don’t have XPS analysis in Pakistan. So, please understand our situation and allow us this time. In future, we will perform XPS analysis for all our experimental samples in advance from foreign countries.

Comment: Figure 1 was not aligned appropriately and not presented according to the methods.

Response: Figure 1 is now aligned appropriately and accordingly and new figure gas been added to the revised manuscript.

Comment: Figure 4 is still not correct.

Response: Figure 4 has been corrected as required and revised figure has been added to the manuscript.

Comment: Figures 10 and 11 are not correlated with each other.

Response: Figures 10 and 11 are revised and modified and ZOI is now extracted and clearly described in the revised manuscript.

Comment: Antifungal activity should be given in the evidential photograph of the plates in the main or supplementary section.

Response: Evidential photographs are now added to the revised manuscript. Thanks for improving the structure and quality of our articles by providing your valuable comments.

Comment: The manuscript contains many flaws in terms of scientific content.

Response: All authors have carefully reviewed the manuscript several times and made necessary changes wherever need and now we hope that our results are well-organized and presented more scientifically.